# EditProp: Consistent Video Style Transfer by Editing Propagation

## Abstract

Video style transfer, which aims to transfer a source video into another video with a different appearance while preserving its original structure, plays an important role in the video production industry. Existing methods often edit the first frame with an image editing tool, and feed it into an image-to-video generation model with source video guidance to generate the edited video. Although such a paradigm enables users to perform creative video editing with powerful image editing tools, it relies heavily on the native propagation capability of the video generation model, which can be limited by having only the first frame as appearance guidance. As a result, the edited video suffers from appearance drifting and structure distortion, leading to severe inconsistencies as time goes on. To this end, we propose *Edit-Prop*, a novel video style transfer framework with two propagation stages: *i)* In the *Keyframe Propagation* stage, the edit in the first keyframe is faithfully propagated to other keyframes with an image-based in-context generation model, producing high-quality edited keyframes with strong appearance consistency. *ii)* Then, in the subsequent *Video Propagation* stage, the source video structure and the propagated keyframes are injected into the video generation model as control signals, providing sufficient appearance and structure guidance to generate the translated video. Experimental results demonstrate that our *EditProp* enables effective transfer to various styles, achieving superior editing results with strong appearance and structure consistency. Furthermore, thanks to our versatile keyframe-based propagation, our framework also enables extra applications such as smooth video style transition and long video style transfer.

## 1 Introduction

Video style transfer aims to translate a source video into another video with different appearances or styles while preserving its original structure. It presents significant application potentials for creative video production in various industries, such as film, education and advertisement. However, achieving video style transfer with traditional video editing tools often requires remarkable financial and human resources, posing significant challenges for both the professionals and the public to create imaginative videos. With the recent advancement of video generation models Kong et al. (2024); Hong et al. (2022); Yang et al. (2024); Wang et al. (2025); HaCohen et al. (2024), it becomes more and more practical to utilize these generative models to facilitate creative video editing and stylization with substantially lower cost.

Existing video style transfer methods can be divided into two main groups: training-free and training-based methods. Training-free methods may require inverting a video to its initial noise, and manipulating the weights in the attention matrices Liu et al. (2023); Qi et al. (2023) or propagating the latent features utilizing temporal correspondence Geyer et al. (2023); Yang et al. (2023). They are often sensitive to hyper-parameters and require extensive case-specific tuning. Early methods adopt Text-to-Image models as basis and would inevitably produce flicker artifacts Qi et al. (2023). On the other hand, training-based methods often utilize an Image-to-Video (I2V) model to generate the video using an edited first frame. They learn a module to inject the source video Zi et al. (2025); Liu et al. (2024b) or its structure signals, such as tracking points Gu et al. (2025), human skeleton Wang et al. (2024) and optical flows Liang et al. (2024) to the video

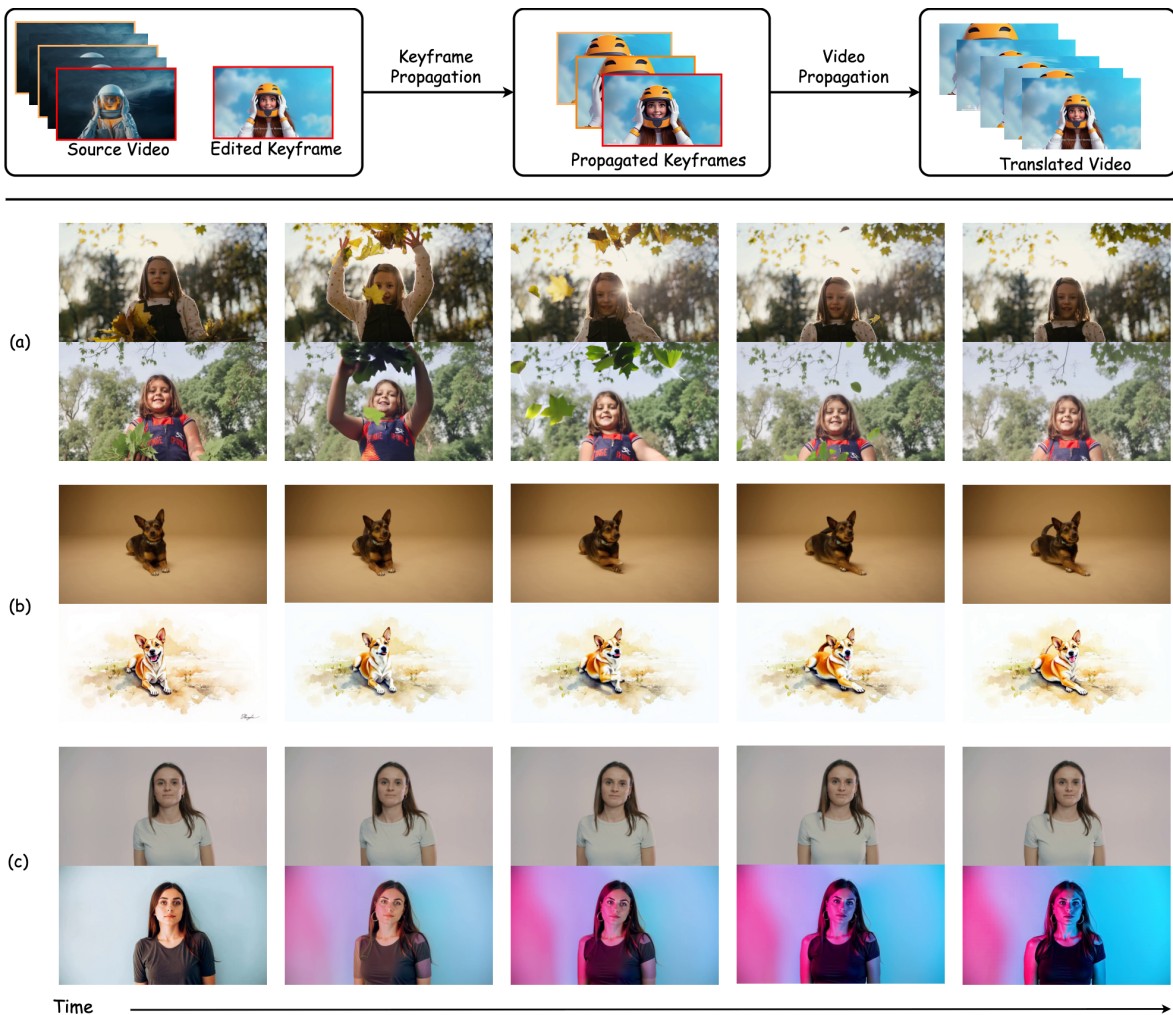

Figure 1: (i) Top: Overview of our framework. Given a source video and one edited keyframe, our framework first propagates the edits in keyframes, obtaining a set of edited keyframes. They are then utilized for video propagation to get the full translated video in a second stage. (ii) Bottom: video style transfer results produced by our method. Our framework enables video translation to various scenarios, such as (a) realistic → realistic style, (b) realistic → watercolor style and (c) realistic → multiple styles with a smooth transition.

generation model to preserve the source video structure. However, with only the first frame as appearance guidance, this paradigm can be limited by the inherent capability of the base I2V model, which struggles at preserving the appearance consistency between the first frame and subsequent frames. In fact, as shown in Figure 2, with only 1 keyframe as guidance, the appearance in the generated frames tends to accumulate deviation as time progresses, resulting in increasing appearance inconsistencies.

In this work, we propose *EditProp*, a novel two-stage framework that achieves both appearance- and structure-consistent video style transfer. In the first *Keyframe Propagation* stage, the edit in the first keyframe is propagated to other keyframes via an image-based generation model, obtaining edited keyframes with high appearance-consistency and quality. This is achieved via an in-context image generation Huang et al. (2024a) model built upon Flux Labs (2024). It takes three images as input conditions: 1) the first keyframe from the source video, 2) another keyframe from the source video and 3) the edited first keyframe. Then, a model is trained to learn the correspondence between the source video keyframes implicitly, and propagate the edits to obtain another edited keyframe. Utilizing the in-context generation capability of the

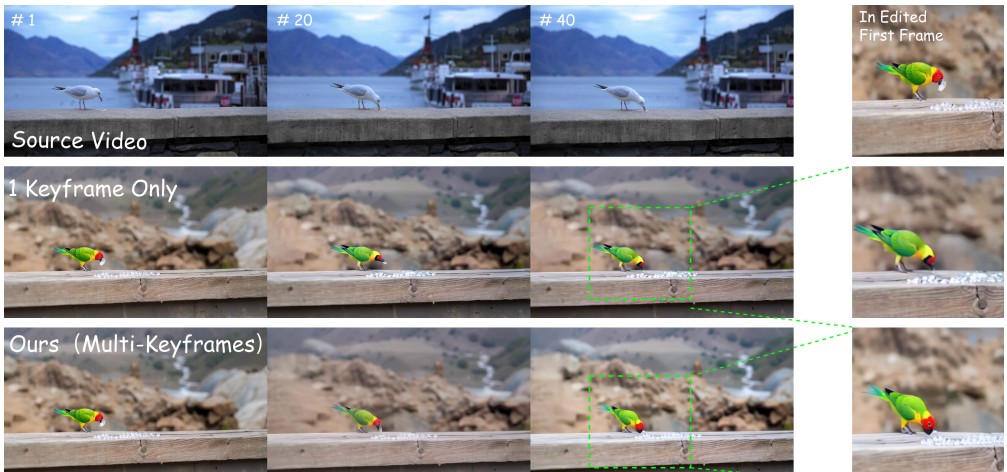

Figure 2: Effect of different numbers of keyframes on editing results. With only 1 keyframe (the middle row), the bird suffers from appearance inconsistency. With multiple edited keyframes (ours), we achieve more consistent editing results.

image generation model Labs (2024); Huang et al. (2024a), we are able to obtain a series of high-quality edited keyframes with high appearance consistency.

Given the edited keyframes in the first stage, the second *Video Propagation* stage aims to propagate the edited keyframes to produce the full translated video following the structure of the source video. This is achieved via *AS-Ctrl*, a two-stream controller that smoothly injects **A**ppearance and **S**tructure signals into the video generation model. In the appearance stream, the propagated keyframes are arranged as a video, with empty frames inserted in-between, ensuring flexible arrangement of keyframe positions. In the structure stream, depth video is selected as the structure representation due to its high precision and expressivity Hu et al. (2025). The condition of the two streams are separately fed into a Diffusion Transformer Jiang et al. (2025); Peebles & Xie (2023), producing intermediate features that are injected into the base model to obtain appearance and structure-consistent video translation results.

We conducted extensive experiments to validate the effectiveness of our *EditProp*. Experimental results demonstrate that our method achieves superior editing results than existing open-source and commercial video style transfer methods, especially in appearance and structure consistency. We also conducted extensive ablation analysis to validate the efficacy of the two stages in the proposed framework. Furthermore, thanks to the strong consistency preservation capability of *EditProp*, our method also opens up new possibilities for extra applications, such as smooth video style transition using edited keyframes of different styles, as shown in Figure 1 (c). To summarize, our contributions are as follows:

- We propose *EditProp*, a novel two-stage framework that achieves video style transfer with strong appearance and structure consistencies.

- In the first *Keyframe Propagation* stage, we design an in-context image generation framework that faithfully propagates the edit in one keyframe to other keyframes, ensuring their appearance consistency and visual quality.

- We design *AS-Ctrl*, a plug-and-play control module that smoothly integrates appearance and structure control signals into the base video generation model.

- Extensive experiments and human studies validated the effectiveness and superiority of the proposed *EditProp*, which also opens up new possibilities for novel applications such as smooth video style transition.

## 2 Related Work

### 2.1 Video Diffusion Models

In recent years, significant advancements have been made in the development of video generation based on diffusion models. Early video generation models Ho et al. (2022); Blattmann et al. (2023); Guo et al. (2024) are directly built upon the text-to-image models Rombach et al. (2021) by injecting trainable temporal layers to model the inter-frame relations. More recent video generation models such as Sora Brooks et al. (2024), CogVideo Yang et al. (2024), HunyuanVideo Kong et al. (2024), and Wan Wang et al. (2025) employ 3D-VAE to compress high-dimensional raw videos into compact latents for efficiency and utilize diffusion transformer with spatio-temporal attention mechanism to generate high-quality results. These models often accept text conditions via joint 3D attention or cross-attention mechanism. They are also adapted to Image-to-Video (I2V) models that generate frames from a condition image. These video models serve as a fundamental backbone for many video tasks, such as video depth estimation Hu et al. (2025) and video editing Zi et al. (2025); Jiang et al. (2025).

### 2.2 Video Style Transfer

Video style transfer aims to generate the desired effects of the target video while preserving some attributes of the source video. Early attempts Liu et al. (2024a); Bao et al. (2023); Geyer et al. (2023); Yang et al. (2023) have utilized training-free frameworks for this task. However, constrained by the pre-trained image diffusion model they employ, these methods may inherit the artifacts and limited generation ability of the basic model and struggle to produce temporally consistent results. Learning-based methods Liang et al. (2024); Gu et al. (2025); Liu et al. (2024b); Zi et al. (2025) provide a more general solution by fine-tuning basic models on large-scale datasets, demonstrating promising results in video style transfer. Specifically, FlowVid Liang et al. (2024) harnesses the benefits of optical flow while handling the imperfection in flow estimation. GenProp Liu et al. (2024b) directly adopts the source video as the control conditions. Diffusion as Shader Gu et al. (2025) leverages 3D tracking videos as control inputs, which could be extracted from source videos to generate target videos for video style transfer Translation. However, these methods primarily rely on the first frame for appearance control, making it challenging to maintain consistency, especially for videos with large motion or long durations. In contrast, our two-stage framework facilitates a smooth and temporally coherent translation by propagating edits in keyframes.

## 3 Methodology

Given a source video $S = \{f_1, f_2, ...f_l\}$ with $l$ frames, $EditProp$ aims to translate it into a video with different appearance or style while preserving its original structure. Following the setup of the previous work Liu et al. (2024b); Gu et al. (2025); Liang et al. (2024), $EditProp$ requires one keyframe to be edited with an external image editing model $\hat{f}_{k_1} = ImageEdit(f_{k_1}), k_1 \in \{1....l\}$. $ImageEdit$ can be any image editing tool, such as FLUX.1-Depth-dev Labs (2024) and Nano-Banana, as long as it preserves the structure of the input keyframe. Leveraging the strong generation and editing capability of advanced image generation models, such a paradigm allows users to edit the keyframe into an arbitrary appearance with any advanced editing tool, offering more flexibility and potential for creative video style transfer translations.

As shown in Fig. 3, given the first edited keyframe, our $EditProp$ framework performs video style transfer in two stages. In the first *Keyframe Propagation* stage, we propagate the edit in the first keyframe $\hat{f}_{k1}$ to other keyframes with in-context learning, obtaining a series of $K$ high-quality and appearance-consistent keyframes. Then, in the second *Video Propagation* stage, we extract the structure $D$ from the source video. Together with the propagated keyframes, the structure information is injected into the base video generation model via *AS-Ctrl*, a two-stream control module that accepts both **A**ppearance and **S**tructure control signals. In the following section, we will elaborate on the *Keyframe Propagation* in Section 3.1 and the *Video Propagation* stage in Section 3.2. Finally, the training and inference process will be presented in Section 3.3.

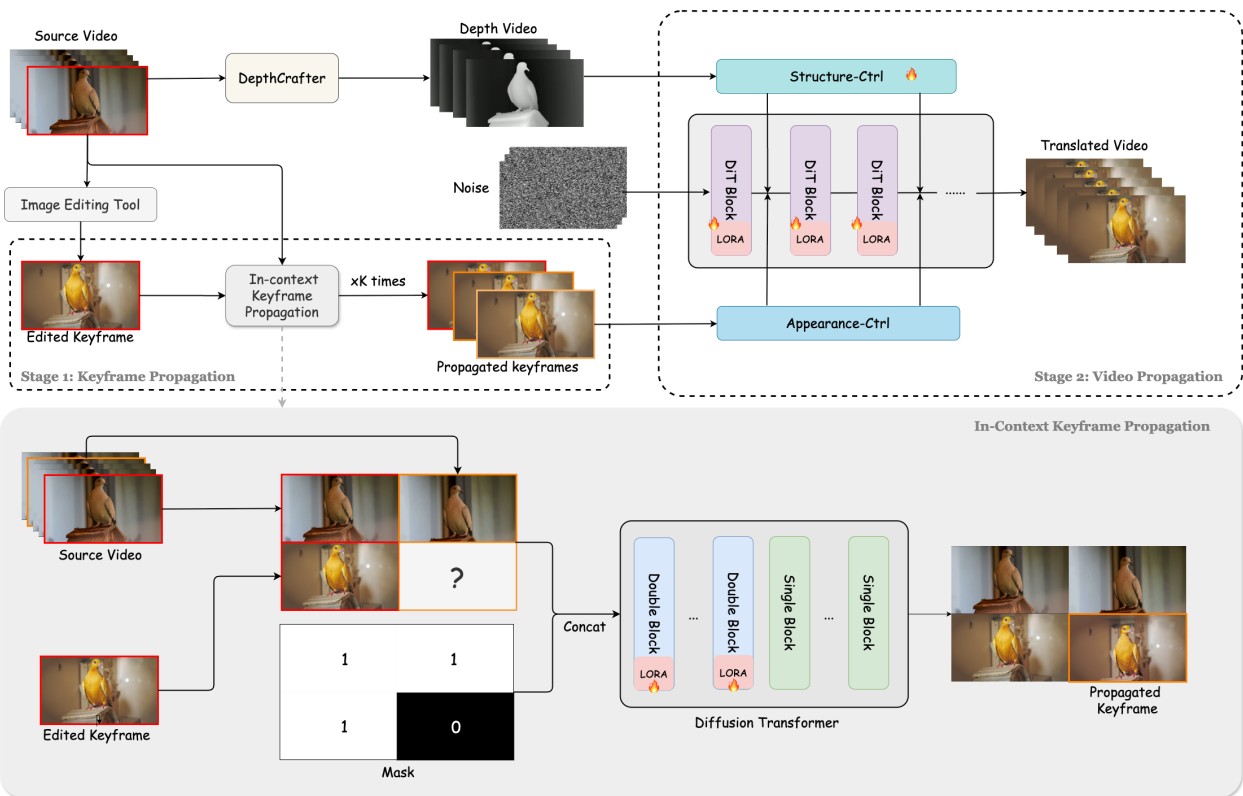

Figure 3: (a) Top: The overall pipeline of our *EditProp* framework. Given a source video and one edited keyframe, we first perform *Keyframe Propagation* in the first stage, obtaining $K$ propagated keyframes. In the second stage, *Video propagation* is performed by taking the appearance and structure (*i.e.*, depth) control signals into the base DiT model. (b) Bottom: Illustration of the proposed in-context keyframe propagation process. Given two keyframes and one edited keyframe, they are concatenated with an empty image, forming a 2x2 panel, with a mask representing the condition regions. They are concatenated and fed into a diffusion transformer Labs (2024), utilizing its in-context generation capability to generate an edited keyframe. Frames with the same timestamps are outlined with the same color.

## 3.1 In-Context Keyframe Propagation

Given the source video and one edited keyframe, previous methods for video style transfer Gu et al. (2025); Liang et al. (2024) extract the source video structure, *e.g.*, depth, and use it as the condition to guide the generation process of a base Image-to-Video (I2V) model. Although such a paradigm can largely preserve the structural consistency in the edited video, the appearance can be twisted and degraded in subsequent frames. In fact, using only the first frame, the appearance of the generated frames degraded significantly as time progresses, leading to distorted and blurry frames, as illustrated in Figure 2.

To this end, other than utilizing a single keyframe, we propose to guide the video generation process with multiple high-quality and appearance-consistent keyframes. Given the source video keyframes and one edited keyframe, we leverage the strong in-context generation capability Huang et al. (2024a) of the recent transformer-based image generation model Labs (2024) to inject contextual information. Keyframe Propagation assumes sufficient visual correspondence between keyframes, which requires scene content and subject identity are maintained across keyframes. The method may degrade for videos containing shot transitions or drastic cross-keyframe appearance changes.

As shown in lower part of Fig. 3, the keyframe propagation model takes three images as input conditions: 1) the first keyframe $f_{k_1}$ in the source video, 2) another keyframe $f_{k_i}$ in the source video and 3) the first

edited keyframe $\hat{f}_{k_1}$. Together with an empty image, they are concatenated into a 2x2 panel. We utilize a binary mask to indicate the regions of generation or condition, so as to utilize the strong in-filling capability of the inpainting model. The process of generating the $i$-th edited keyframe $\hat{f}_{k_i}$ is as follows:

$$\hat{f}_{k_i} = P \left( \begin{bmatrix} f_{k_1} & f_{k_i} \\ \hat{f}_{k_1} & 0 \end{bmatrix} ; \begin{bmatrix} 1 & 1 \\ 1 & 0 \end{bmatrix} \right). \tag{1}$$

Thanks to the powerful in-context generation capability of diffusion transformer combined with the inpainting formulation, we could achieve this objective by training only a lightweight LoRA Hu et al. (2022) on the model. As show in Fig. 4, during inference, Eq. 1 can be performed for multiple times to generate a certain number of edited keyframes. These propagated keyframes are then utilized in the second stage to generate appearance-consistent videos.

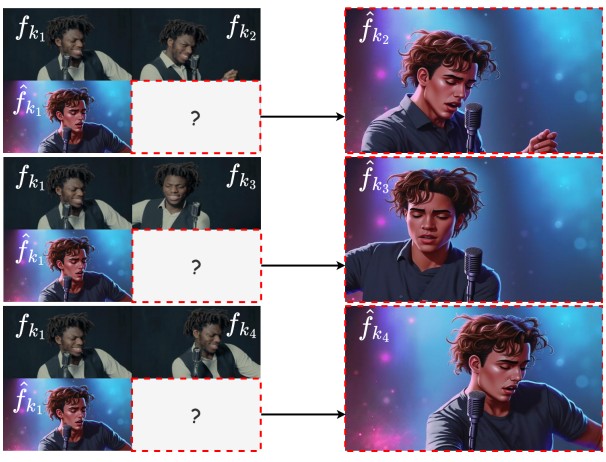

Trained on high-quality paired videos Zi et al. (2025), our keyframe propagation model learns to establish robust correspondences between temporally distant keyframes, mitigating the appearance drift in the single-keyframe-controlled video editing and contributes to long video editing (see Figure ??).

Figure 4: Illustration and results of our keyframe propagation process to generate multiple keyframes.

### 3.2 Appearance-Structure Control for Video Propagation

Given the edited keyframes, we aim to utilize a video generation model to propagate the edited keyframes into a full video while following the source video structure. In this work, we choose video depth Hu et al. (2025) as the structure representations due to its high precision and expressivity. However, it is nontrivial to simultaneously inject keyframe and depth signals into a video generation model due to their distinct characteristics and effects range. Previous work has explored utilizing a single keyframe Guo et al. (2023); Feng et al. (2023), depth Peng et al. (2024); Lin et al. (2024) or one single frame with depth as conditions Feng et al. (2023), but jointly utilizing multiple keyframes and depth as control conditions under the DiT architecture remains unexplored.

To this end, we propose *AS-Ctrl*, a two-stream module that smoothly integrates appearance and structure signals into the video generation model. *AS-Ctrl* has two branches, one accepts video depth as control signals and the other takes keyframes as input. Following the flexible design of VACE Jiang et al. (2025), we utilize video and mask to represent each condition and their temporal positions. For the structure branch, the input is the video depth extracted by DepthCrafter Hu et al. (2025), accompanied by an empty mask of the same size, indicating the existence of the depth signals at each temporal position:

$$I_S = [S; M_S], \tag{2}$$

where [;] indicates concatenation along the channel dimension, $S$ represents the video depth and $M_S$ is an empty mask with all zeros.

For the appearance branch, the propagated keyframes are arranged into a video that has the same number of frames with source video, with positions without keyframe filled with zero.

$$I_A = [A; M_A], \tag{3}$$

where $A = \{\hat{f}_{k_1}, 0, ..., \hat{f}_{k_i}, 0, ..., \hat{f}_{k_n}\}$ and $M_K = \{1, 0, ..., 1, 0, ..., 1\}$, with 1 indicating the availability of keyframes and 0 otherwise. The two inputs are fed into their corresponding controllers, obtaining structure and appearance control signals, respectively.

$$\begin{aligned} c_s &= \texttt{S-Ctrl}(I_S) \\ c_a &= \texttt{A-Ctrl}(I_A) \end{aligned} \tag{4}$$

They are then added to the base video generation model with addition:

$$c_{base}^n \leftarrow c_{base}^n + \alpha_s * c_s^n + \alpha_a * c_a^n, \tag{5}$$

where $c_{base}^n$ represents the feature of the base model at the $n$-th block. $\alpha_s$ and $\alpha_a$ are scalars controlling the strength of the two control signals, respectively. In practice, we inject the control signals into the base model for every other block. The two-stream structure enables users to independently control the strength of the two signals, offering more flexibility to dynamically adjust them.

In contrast to existing architectures Feng et al. (2023); Jiang et al. (2025), which do not simultaneously support multiple keyframes and structural control signals, our framework dynamically accepts both appearance and structure guidance in a more flexible and adjustable manner, making it better suited for following the appearance conditions generated by the keyframe propagation stage.

### 3.3 Implementation Details

**Keyframe Propagation.** The in-context keyframe propagation model $P$ is trained on paired videos selected from the Senorita-2M Zi et al. (2025) dataset. We utilized the style transfer split only since it is highly aligned with the objective of our task. We empirically found that jointly utilizing other splits does not improve the performance. It is based on the inpainting version of FLUX with a LoRA of rank 64. The training process takes a total of 50k steps with a learning rate of 1e-4 at 480×832 resolution. During training, $k \in [1, 6]$ keyframes are randomly sampled from the source video. For inference, the keyframes are uniformly sampled. Given one edited keyframe, we perform keyframe propagation for a total of $k - 1$ times, each generating one edited keyframe. We follow the default setting of Flux-fill and utilize a guidance scale of 30.

**Video Propagation.** The training of AS-Ctrl is conducted on a high-quality video dataset, with 65K videos extracted from Koala-36M, each accompanied with a video depth extracted by DepthCrafter Hu et al. (2025). Wan-1.3B Wang et al. (2025) is selected as the base video generation model, with both streams of *AS-Ctrl* initialized from the control block of VACE Jiang et al. (2025). During training, we randomly sampled 1 to 6 keyframes with corresponding masks as the conditions. The model was trained on videos with a resolution of 480×832. For the first 5k steps, we use videos of 41 frames for training and then 81-frame videos are utilized for training the next 3k steps. The keyframe control branch is frozen to maintain its original keyframe-based generation capabilities while the depth branch is fully-finetuned. A lora of rank 32 is also tuned on the base video generation model to better accommodate the two control conditions.

## 4 Experiments

### 4.1 Benchmark Design

To evaluate our model, we curate KP-Bench, a single-shot benchmark specifically designed for evaluating video style transfer translation with varying motion degrees.[1] Compared with existing benchmarks Wu et al. (2023); Liu et al. (2024a), it contains videos of various difficulties, covering different aspect ratios, motion degrees, target styles that are more suitable for evaluating recent video generation models. It contains a total of 40 high-quality videos, each accompanied with a target prompt describing the translated video. Each video was edited into 3 randomly selected styles, such as cyberpunk, watercolor, and 3D animation. For each video and target style, we edit the first frame of the video with FLUX-dev-depth[2]. We consider two types of metrics to evaluate the quality of the translated video.

- Appearance Consistency and Structure Consistency. The former is evaluated by computing the similarity between the generated frames and the first edited frame, which is then averaged across all frames in the generated video. For structure consistency, we extracted the video depth of the source and translated video, and computed the relative absolute error between them.

---

[1]We also performed evaluation on other video style transfer benchmarks such as TGVE Wu et al. (2023) and DAVIS-Edit Liu et al. (2024a) and put them into supplementary material due to space limit. We strongly encourage readers to refer to our supplementary material for more results.

[2]https://huggingface.co/XLabs-AI/flux-controlnet-depth-v3

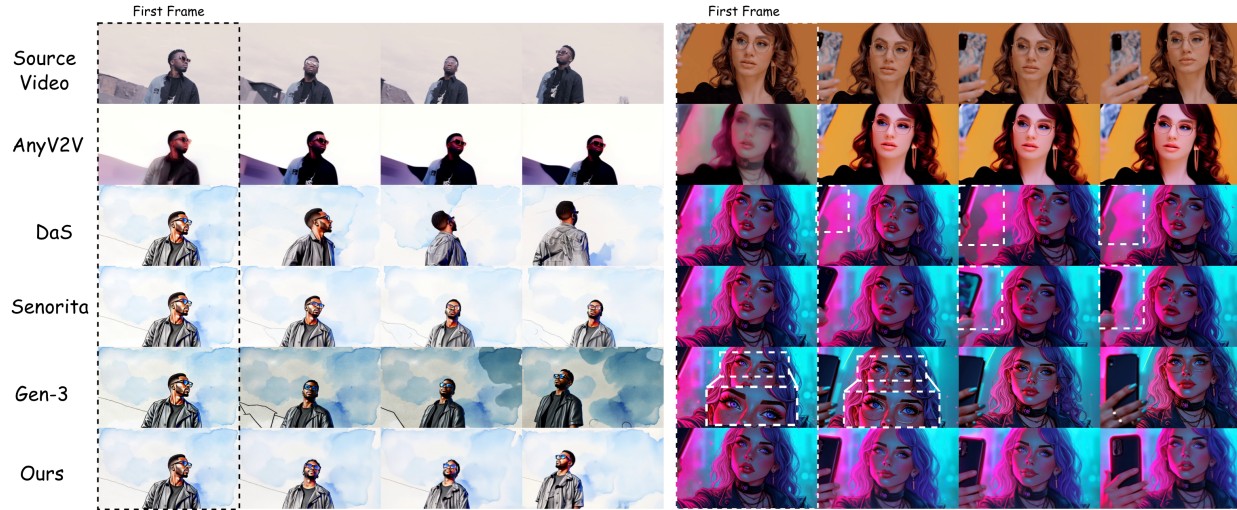

Figure 5: Qualitative comparison between our method and other approaches, with the first frame indicated by the dashed rectangle. Our method achieves the best appearance consistency between edited frames and structure consistency with the source video.

| Method | Appearance Consistency | Structure Consistency | Visual Quality |
|---|---|---|---|
| AnyV2V Ku et al. (2024) | 0.0% | 2.36% | 1.57% |
| DaS Gu et al. (2025) | 10.46% | 8.13% | 15.32% |
| Senorita Zi et al. (2025) | 7.64% | 8.32% | 9.54% |
| Runway/Gen-3 | 24.60% | 32.56% | **48.43%** |
| Ours | **57.30%** | **48.63%** | 25.14% |

Table 1: Human evaluation results. The numbers represent the rate that the method is preferred over others.

- Quality: We further adopt two metrics from V-Bench Huang et al. (2024b) to measure the temporal coherence of the generated video including motion flickering and motion smoothness.

## 4.2 Comparison with video style transfer Methods

**Baselines.** We compared our method with existing open-source video editing methods, including 1) the inversion-based method AnyV2V Ku et al. (2024), 2) DaS Gu et al. (2025), a first-frame-guided editing method based on the tracking point video extracted from the source video. 3) instruction-based editing method, Senorita Zi et al. (2025)[3], which takes the source video and the text instruction as input. Besides, we also compared with the restylization function of Gen-3 from Runway, which also takes the source video and the first edited frame as input. To comprehensively evaluate these methods, we conduct human, quantitative, and qualitative assessments.

**Human Evaluation.** Since there are no ground truth videos as reference, we mainly adopt a user study for evaluation. We performed a human evaluation to compare our method with baseline methods. Besides KP-Bench, we also include samples from existing benchmarks such as TGVE-Bench Wu et al. (2023) and DAVIS-Bench Ku et al. (2024). Finally a total of 60 groups of samples are provided to 86 participants. The participants consists of lab members and crowdsourcing workers. For each test case, participants were presented with 5 videos (one per method) corresponding to the same source video and style, and asked to independently select the best-performing video on each criterion (visual quality, appearance consistency, structure consistency). The evaluation results are shown in Table 1, which shows that our method demon-

---

[3]Although this method is also trained on the Senorita-2M dataset, we utilized it to train the image-level propagation in stage 1 only.

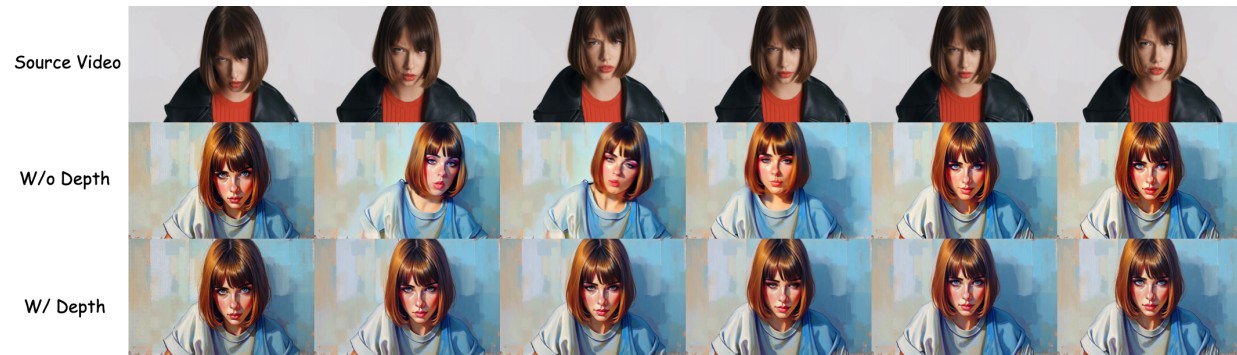

Figure 6: Effect of depth control for video editing. Both editing results take the same number of keyframes as input. Although the start and end positions of the person can be controlled via keyframes, the structure in intermediate frames can deviate from the source video without depth control.

strates the highest appearance consistency and structure consistency. On the other hand, it demonstrates the second-best video quality, and is only inferior to the propriety model Gen-3.

| Method | Appearance Consistency ↑ | Structure Divergence ↓ | Motion Flickering↑ | Motion Smoothness ↑ |
|---|---|---|---|---|
| AnyV2V Ku et al. (2024) | 0.7030 | 1.2453 | 0.9816 | 0.9873 |
| DaS Gu et al. (2025) | 0.9149 | 1.1127 | 0.9831 | **0.9903** |
| Senorita Zi et al. (2025) | 0.9157 | 0.8505 | 0.9831 | 0.9901 |
| Runway Gen-3 | 0.8882 | 1.0674 | **0.9839** | 0.9883 |
| Ours | **0.9213** | **0.7836** | 0.9833 | **0.9903** |

Table 2: Quantitative Evaluation on KP-Bench. Our EditProp demonstrates the best appearance consistency, structure consistency and motion smoothness.

**Quantitative Comparison.** The quantitative results of objective metrics are shown in Table 2. Our method achieves the highest appearance consistency and the lowest structure divergence among all methods, demonstrating the advantage of keyframe propagation and structure control. Besides, our method also exhibits the best motion smoothness, showing that our method can also produce temporally coherent results.

**Qualitative Comparison.** We qualitatively compared our approach with baselines. Two examples are illustrated in Figure 5, where the first frame is indicated in the dashed rectangle box. We observed that AnyV2V produces blurry and inconsistent results even on the first frame, showing that the inversion-based method is case-sensitive and less robust. On the other hand, DaS exhibits limited structure preservation and motion transfer capability. For example, in the first example of fig:qualit, the person turns around in a distinct direction from the source video in DaS. Besides, the results produced by Senorita show limited motion following capability, while appearance distortions are also observed. Finally, although Gen-3 often produces results with high visual quality, it struggles to maintain consistency with the edited first frame. For example, in the fifth row, the results in the left column exhibit a different tone with the first frame. In the right column, the girl was not wearing glasses in the first frame, but the glasses unexpectedly showed up in later frames, which can be leaked from the source video.

## 4.3 Ablation Analysis

**Necessity of the Two-Stage Design.** We analyze the contribution of each stage to the final editing quality. Using only Stage 1, we rely solely on our keyframe propagation model to propagate the edit to all frames and concatenate them into a video. Using only Stage 2, we perform single-image-to-video generation with depth-based structural control. As shown in Table 3, both stages are essential for high-quality results. Stage 1 alone yields videos with strong appearance consistency but poor structural coherence, as it lacks explicit control over scene structure. Conversely, Stage 2 alone improves temporal and structural consistency

| Method | Appearance Consistency ↑ | Structure Divergence ↓ | Motion Flickering↑ | Motion Smoothness ↑ | Time Cost(s) |
|---|---|---|---|---|---|
| stage 1 only | **0.9294** | 0.9120 | 0.9819 | 0.9884 | 2046 |
| stage 2 only | 0.9102 | 0.7840 | 0.9823 | 0.9901 | 42 |
| stage1 + stage2 (joint trained) | 0.9145 | 0.7866 | 0.9830 | 0.9901 | 156 |
| Ours (frozen apperance branch) | 0.9213 | **0.7836** | **0.9833** | **0.9903** | 156 |

Table 3: Study on the effect of two stages.

but suffers from degraded appearance fidelity due to the absence of keyframe-guided propagation. By jointly leveraging both stages, we achieve a balanced solution that excels in both structural alignment and visual consistency.

**Effect of control conditions.** We also qualitatively compared how the number of keyframes and the video depth condition affect the editing results in Figure 2. With only 1 keyframe used for generating the edited video, the bird suffers from significant ID distortions. With 2 keyframes, the editing quality gets noticeable improvement, yet some details of the main subject are still unexpectedly altered. Finally, with all 4 keyframes used, we achieve high-quality editing results with high appearance consistency. Besides, we visualize how depth control affects the editing results in Figure 6. Although using keyframes can somehow control the position of the person, its pose and structure in intermediate frames are not controllable without depth control.

**The advantage of depth control.** We further study how the choice of depth map affects the editing results. Besides depth, we also extracted semantic segmentation map, optical flow and bounding box from the source video, and train a seperate model under the same protocol for each condition. The comparison is shown in Table. 4. It can be observed that, although these menthods results in similar appearance consistency and motion smoothness, they differ from depth in structure divergence. These results show the advantage of depth over other conditions in the video style transfer task.

| Control Condition | Appearance Consistency ↑ | Structure Divergence ↓ | Motion Flickering↑ | Motion Smoothness ↑ |
|---|---|---|---|---|
| Semnatic Map | 0.9199 | 0.8245 | **0.9834** | 0.9900 |
| Optical Flow | 0.9210 | 0.8023 | 0.9822 | 0.9898 |
| Bounding Box | 0.9135 | 0.8325 | 0.9754 | 0.9865 |
| Ours | **0.9213** | **0.7836** | 0.9833 | **0.9903** |

Table 4: Comparison of different control conditions.

**Effect of keyframes.** We investigated how the control signals influence the quality of the reconstructed videos. We randomly selected 100 videos from the validation set to evaluate the reconstruction quality with different numbers of keyframes and availability of video depth. The results are shown in Figure 7. Note that we will remove the first stage keyframe propagation when using only one frame. It can be observed that, utilizing depth signals can significantly improve the reconstruction results, resulting in significantly higher PSNR and SSIM. On the other hand, increasing the number of keyframes also improves reconstruction quality, which saturates with about 4 keyframes. This result shows that the keyframe and depth signals are both necessary for high-quality video reconstruction and are complementary to each other.

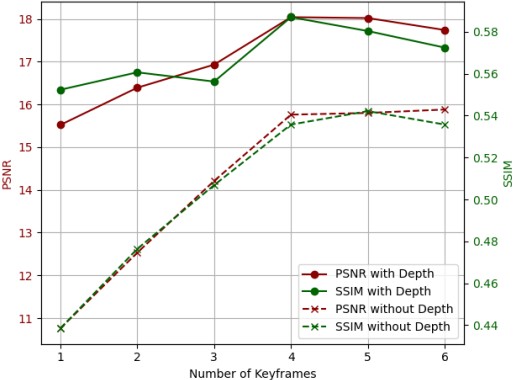

Figure 7: Video reconstruction quality with different numbers of keyframes and the effect of video depth condition for the results.

**Performance Changes with Various Inter-Keyframe Difference.** To further investigate the performance sensitivity of our method with various inter-keyframe visual differences, we further group all the source videos in our KP-Bench according to the CLIP similarity between the first and last keyframe. The quantitative performance of our method with different inter-keyframe difference is shown in Table. 5. It

can be observed that, our method performs reasonably good when the inter-keyframe difference is small. However, when the inter-keyframe difference increases, the appearance consistency drops significantly.

|  | Appearance | Structure |
|---|---|---|
| Inter-Keyframe Similarity | Consistency | Divergence |
| 0.9 - 1.0 | **0.9322** | **0.7724** |
| 0.8 - 0.9 | 0.9292 | 0.7883 |
| < 0.8 | 0.9102 | 0.8023 |

Table 5: Performance degrades with increasing inter-keyframe visual difference.

## 5 Discussion

### 5.1 Conclusion

In this work, we propose *EditProp*, a novel two-stage framework that achieves video style transfer with high appearance and structure consistency. The first stage performs keyframe propagation, which translates the edit in one keyframe to other keyframes. The second video propagation stage generates translated video with appearance guidance from the first stage. Our method achieves superior editing results than other methods, enabling interesting applications such as video style transition and long video style transfer.

### 5.2 Limitations

Our method can be limited by the inherent capability of the base generation model, which can generate poor human bodies and glyphs. Besides, when there are limited or no correspondence between keyframes, our method may fail to perform keyframe propagation. Also, the capacity of our method relies heavily on the quality of the paired video dataset, which be further improved with higher-quality dataset. On the other hand, our evaluation covers only shot-level video editing, which could be further extended to the realm of multi-shot video style transfer.

### 5.3 Future Work

In this work, we explored only global video editing, *i.e.*, style transfer. It would be interesting to explore other types of video editing using our proposed paradigm. Besides, the keyframe-based editing strategy enables potential applications such as video style transfer. which can be achieved by transfering different keyframes into different styles and propagate them in a video. Also, our method also provides the potential to performance long video style transfer by selecting multiple keyframes in a long video and run our method in each segment.

## Broder Impact

As a video editing technique, video style transfer can be potentially misused to generate deceptive or manipulated content. Since our method relies on the first edited keyframe, an ethical check on the first edited keyframe could help limit the potential for misuse.

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
