# OpenReview forum: "EditProp: Consistent Video Style Transfer by Editing Propagation"
_TMLR — Under review for TMLR_

### Review · Reviewer_whQr · 2026-04-02

**Summary Of Contributions:**

This paper proposes an algorithm for video style transfer focused on preserving structural and appearance consistency. The motivation comes from the observation that existing methods primarily operates by editing a single keyframe toward the given style (the first one) and generating the subsequent sequence, thus vulnerable to any style shifts in subsequent frames. The idea is to generate multiple keyframes with desired styles, via LoRA-based in-context "keyframe propagation" mechanism, and then condition video generation on these multiple frames. In addition to this "appearance-stream" conditioning, the method also uses "structure-stream" conditioning as well.

In my opinion, the key strengths are:
- The limitation of prior approaches noted in the paper seems interesting and is worth sharing with the community.
- The keyframe propagation step is quite reasonably designed. It effectively circumvents the difficulty of generating multiple keyframes with consistent styles.
- The construction of a new benchmark is meaningful, and will be useful for future works.

The key weaknesses are:
- **Weak validation.** The comparison is mainly conducted through human evaluation, which is limited in terms of the scale and breadth. In terms of scale, 30 videos rated by 37 participants seems that it is not the scale which could guarantee generalizability of the method, especially given that no details on how experiment subjects have been recruited and guided is provided. In terms of the breadth, the comparison is mainly done on the newly constructed dataset---KP-Bench. The results on other datasets are only qualitative, and lacks comparisons against baselines.
- **Weak ablations.** One of the technical contributions seem to be the AS-Ctrl, which applies appearance and structural control signals under a separate pipeline. Ablations on each pipeline will help us better understand the role of each controlling unit.
- **Somewhat naïve keyframe selection.** This is a minor concern, but how the keyframes are selected seems a bit naïve, especially given that having multiple keyframes is one of the key differences of the proposed method. There are ablations backing up the choice of the number of frames, but I am not sure why "random" sampling will be an effective choice; in the worst case, they could be selected to be very temporally close to each other. The paper will benefit from comparing with additional simple baselines, e.g., uniform sampling (i.e., choosing equidistant frames).

**Audience:**

Yes

**Audience Explanation:**

The limitation of the previous work the paper describes is interesting by itself. The core idea itself is very reasonable as well. No matter how concretely the research was executed, these two aspects (limitations, idea) are worth sharing.

**Broader Impact Concerns:**

I do not see one, but I do not think it should be 100% required for this paper to have such dedicated section. It is recommendable to include one, as video style transfer has a possibility to be used for deepfake, but should be optional.

**Claims And Evidence:**

No

**Claims Explanation:**

As described above, the paper will benefit by incorporating a few of the following suggestions:
- increasing the number of experimental subjects,
- increasing the number of videos,
- increased number of out-of-KP qualitative examples (or better, quantitative comparisons), compared against baselines
- extended discussions on the human evaluation protocol

(However, I do understand that the former two may be quite difficult to pull off given the limited time).

**Requested Changes:**

- Better evaluation of the proposed method, as described above
- Ablations on the building blocks of AS-Ctrl, with any further discussions on its novelty.
- Further justifications on the keyframe selection mechanisms, preferrably with experimental backups.

---

> ### Author Response · Authors · 2026-06-22
> **Response to Reviewer whQr**
>
> We thank Reviewer whQr for the balanced assessment and constructive suggestions.
>
> ### Keyframe Selection
>
> Our inference uses **uniform sampling** (equidistant keyframes), not random sampling. Section 3.3 contained a writing error that stated "randomly sampled" for inference; the Future Work section's reference to "uniform keyframe sampling strategy" correctly describes our implementation. We have corrected this in our revision. We also provide a comparison between uniform sampling and random sampling strategy as follows.
> | Sampling Strategy | Appearance Consistency ↑ | Structure Divergence ↓ |
> | --- | --- | --- |
> | Random | 0.9106 | 0.7944 |
> | Uniform (Ours)| 0.9213 | 0.7836 |
>
>
> ### Human Evaluation Scale and Breadth
>
> Thanks for the suggestion. We further extand the human study to 86 participants and 60 groups of samples. The participants consists of lab mambers and crowdsourcing workers. While the videos include the results in all of KP-Bench, DAVIS-Edit and TGVE Benchmark.
> The results are as follows and are updated in Table 1 in our revision, where our method still achieves best performance across all metrics on both benchmarks, consistent with the KP-Bench results.
> | Method | Appearance Consistency | Structure Consistency | Visual Quality |
> | --- | --- | --- | --- |
> | AnyV2V | 0.0% | 2.36% | 1.57% |
> | DaS | 10.46% | 8.13% | 15.32% |
> | Senorita | 7.64% | 8.32% | 9.54% |
> | Runway/Gen-3 | 24.60% | 32.56% | **48.43%** |
> | Ours | **57.30%** | **48.63%** | 25.14% |
>
> ### Ablations on the AS-Ctrl
> We compared i) the effectiveness of both stage 1 and stage 2 ; and ii) the training stragey in Table 3 in our revision. The results are as follows. It can be observed that both stages are essential while the proposed frozen training strategy provides better results than joint training.
> | Method | Appearance Consistency ↑ | Structure Divergence ↓ | Motion Flickering ↑ | Motion Smoothness ↑ |
> | --- | --- | --- | --- | --- |
> | stage 1 only | **0.9294** | 0.9120 | 0.9819 | 0.9884 |
> | stage 2 only | 0.9102 | 0.7840 | 0.9823 | 0.9901 |
> | stage1 + stage2 (joint trained) | 0.9145 | 0.7866 | 0.9830 | 0.9901 |
> | Ours (frozen appearance branch) | 0.9213 | **0.7836** | **0.9833** | **0.9903** |
>
> We also compared depth with several typical control conditions, including semantic map, optical flow and bounding box. Of all these conditions, depth provides best structure consistency and appearance consistency.
>
> | Control Condition | Appearance Consistency ↑ | Structure Divergence ↓ | Motion Flickering ↑ | Motion Smoothness ↑ |
> | --- | --- | --- | --- | --- |
> | Semantic Map | 0.9199 | 0.8245 | **0.9834** | 0.9900 |
> | Optical Flow | 0.9210 | 0.8023 | 0.9822 | 0.9898 |
> | Bounding Box | 0.9135 | 0.8325 | 0.9754 | 0.9865 |
> | Ours (Depth) | **0.9213** | **0.7836** | 0.9833 | **0.9903** |
>
>
> ### Keyframe Selection Strategy
> **Sampling Strategy:** We compared our uniform sampling with random sampling during inference (averaged across 5 times). It can be observed that uniform sampling provides a simple but effective strategy.
> | Sampling Strategy | Appearance Consistency ↑ | Structure Divergence ↓ |
> | --- | --- | --- |
> | Random | 0.9106 | 0.7944 |
> | Uniform (Ours)| 0.9213 | 0.7836 |

---

### Review · Reviewer_A682 · 2026-04-06

**Summary Of Contributions:**

## Summary
This paper explores video style transfer, where a user makes a change to one keyframe and wants that change to spread throughout the video while keeping the overall structure intact. The work suggests that older image-to-video pipelines that use the first frame as a guide often lose their look over time because the video model only sees one appearance reference. To fix this, they suggest a two-stage approach: first, an image-based model that understands its surroundings spreads the edit from one keyframe to several others; second, a video generator uses both these spread keyframes and the depth information from the original video as control signals through a two-stream controller called AS-Ctrl. The paper says that the appearance is more consistent and the structure is better preserved than many other methods, and it also mentions that it could be used for transferring videos over long periods and making style transitions smoother.


## Strengths
1. The work is well motivated. The core intuition is clean and easy to understand: if a single reference frame leads to appearance inconsistency and performance decay, we should use multiple key frames as references.
2. The proposed two-stage method is also well-aligned with the motivations of the work.
3. The evaluation results over the entire pipeline over different benchmarks demonstrates competitive results both qualitatively and quantitatively.

## Shortcomings
My biggest concerns for this work is that the proposed framework consists of many different independent components and each component could have different variabilities but they were not fully studied in this work. I can provide a few examples:
1. The work uses depth maps as the structural signal of frames. Is depth map the only choice for structural maps? Can we use bounding boxes or semantic maps? Why or why not?
2. The keyframe propagation process based on in-context guidance is fundamental to the success of stage 2. However, it is not fully studied in the ablation study of the work. Does it rely on specific choice of diffusion models? How good is the quality of the transferred key frames? Does increasing the number of contextual pairs further improve the performance?
3. Does Stage 2 rely on specific choices of base video generation model and control block, which currently are Wan-1.3B and VACE?

In addition, the author claims that the model trained on internal datasets and the details of computing appearance consistency and structure consistency in the curated KP-Bench are missing, which leave the reproducibility of the work questionable.

**Audience:**

Yes

**Audience Explanation:**

Even though video style transfer based on deep learning has been studied for almost a decade, controllable video generation in general has been a hot topic recently due to latest advancement of genAI technologies like diffusion model and video style transfer could be viewed as a sub-topic of controllable video generation.

**Broader Impact Concerns:**

As a methodology work that make improvements to existing works on a well-established task, I don't think there's any broader impact concerns specifically about this work.

**Claims And Evidence:**

Yes

**Claims Explanation:**

1. The core intuition of the work is well supported by a concrete implementation of in Section 3 of the method and both qualitative comparison and quantitative ablation study results.
2. There's a lack of thorough investigation into individual components of the proposed framework and the justification of specific choice made by the authors as I suggested in the shortcomings section. I would deem this part partially supported.
3. The claimed superior performance based on quantitative evaluation results over the entire pipeline is also partially supported due to lack of details on the metrics and datasets.

**Requested Changes:**

1. In the Figure 1, it is hard for the reviewer to spot where is the appearance inconsistency of the bird. Can the author elaborate more on where are the appearance inconsistencies?
2. I suggest the author identify each individual component that could be replaced by a different variability in the entire proposed pipeline. I don't think asking for thorough investigation into each individual component is necessary or reasonable but I highly encourage the author to provide some justifications for the current choice of each individual component of the work. It would be better if the author could provide some qualitative or quantitative study results by varying the choices on at least some of the important components.
3. Long video style transfer and video style transition look like interesting applications of the proposed framework. It would make the work more solid if the author could provide more qualitative examples and comparisons against baseline methods. Otherwise, I would suggest the authors move it to discussions as future works,
4. There are a few typos in the work. In the second bullet point appearing in Page 3, visual qualty - visual quality. At the bottom of page 4 Keyframe Propgation -> Keyframe Propagation. In Section 3.2, each conditions -> each condition.

---

> ### Author Response · Authors · 2026-06-22
> **Response to Reviewer A682 [Requested Changes]**
>
> We thank Reviewer A682 for the insightful and constructive feedback. Our responses are as follows.
>
> ---
>
> ### 1. Figure 1 Bird Appearance Inconsistency
>
> We acknowledge this is not immediately apparent in the current figure layout. In Figure 2 (the more detailed ablation figure showing the bird example), the inconsistencies are visible as: (1) the bird's plumage color and texture shifts in later frames compared to the first edited frame; (2) with only 1 keyframe guidance, the bird's beak detail changes in frames 3.
>
> ---
>
> ### 2. Component Choices Justification and Ablation
>
> We provide brief justifications for each major design choice, and will add these to the paper:
>
> | Component | Current Choice | Rationale |
> |---|---|---|
> | Base model for Stage 1 | FLUX (DiT, inpainting) | Strongest open-source image DiT with in-context LoRA capability; SD-based models lack this architecture |
> | Base model for Stage 2 | Wan-1.3B | Strong open-source video DiT; AS-Ctrl is designed to be plug-and-play with other video DiT models |
> | Control block | VACE | Strong, open-source reference architecture; AS-Ctrl modifies it for dual-stream operation |
> | Structure signal | Depth (DepthCrafter) | Temporally consistent, style-agnostic, geometrically precise |
> | Keyframe sampling | Uniform at inference | Avoids temporal clustering; random during training for generalization |
>
> We also provide some quantitative comparisons to justify the design of our method as follows:
>
> **Base Model for Stage 2:** Our method is compatible with other base models such as Wan-14B, which achieves even better performance.
> | Model | Appearance Consistency | Structure Divergence|
> | --- | --- | --- |
> | Wan-1.3B | 0.9322 | 0.7724 |
> | Wan-14B  | 0.9356 | 0.7801 |
>
> **Control Conditions:** We compared depth control with semantic map, bounding box and optical flow, the results show that depth provides the best structure similarity than other control conditions. This experiment is also included in Table 4 in our revision.
> | Control Condition | Appearance Consistency ↑ | Structure Divergence ↓ |
> | --- | --- | --- |
> | Semantic Map | 0.9199 | 0.8245 |
> | Optical Flow | 0.9210 | 0.8023 |
> | Bounding Box | 0.9135 | 0.8325 |
> | Ours (Depth) | **0.9213** | **0.7836** |
>
> **Sampling Strategy:** We compared our uniform sampling with random sampling during inference (averaged across 5 times). It can be observed that uniform sampling provides a simple but effective strategy.
> | Sampling Strategy | Appearance Consistency ↑ | Structure Divergence ↓ |
> | --- | --- | --- |
> | Random | 0.9106 | 0.7944 |
> | Uniform (Ours)| 0.9213 | 0.7836 |
>
> ---
>
> ### 3. Long Video Transfer and Style Transition
>
> We agree these applications currently lack quantitative baselines. We thus move them to the discussion of future work. We hope to systematically study this task in the future.
>
>
> ---
>
> ### 4. Typos
>
> Thank you for pointing these out. We have corrected these typos in our revision.
>
> ---
>
> ### Short Comings -- Internal Dataset Reproducibility
> The in-context keyframe propagation model trained on paired videos selected from the Senorita-2M dataset.  We utilized the style transfer split only since it is highly aligned with the objective of our task. We empirically found that jointly utilizing other splits does not improve the performance. The training of AS-Ctrl is conducted on a high-quality video dataset, with 65K videos extracted from Koala-36M, each accompanied with a video depth extracted by DepthCrafter. We have included these in the Section 3.3 in our revision.

---

### Review · Reviewer_Zjkk · 2026-06-08

**Summary Of Contributions:**

This paper proposes **EditProp**, a two-stage video style transfer framework that addresses the appearance drifting problem in existing methods, which rely solely on the first edited frame as appearance guidance. The first stage (**Keyframe Propagation**) uses an in-context image generation model built on FLUX to propagate the edit from one keyframe to multiple keyframes via a lightweight LoRA. The second stage (**Video Propagation**) introduces **AS-Ctrl**, a dual-stream controller that simultaneously injects appearance signals (propagated keyframes) and structural signals (depth maps) into a base video generation model (Wan-1.3B).

**Paper strengths:**
- The two-stage design is well-motivated with clear division of responsibilities between appearance and structure control.
- The keyframe propagation mechanism is lightweight (LoRA only) and effectively leverages the in-context generation capability of FLUX.
- Ablation studies systematically validate each major design choice (Table 3, Figures 2, 6, 7).
- Consistent improvements are demonstrated across three benchmarks (KP-Bench, TGVE, DAVIS-Edit).

**Paper weaknesses:**
- The source and quality of edited GT keyframes used for LoRA training are not clarified.
- The method's implicit assumption about inter-keyframe visual correspondence is not explicitly stated, which may lead readers to overestimate the method's applicability.
- There is a large unexplained gap between quantitative metrics and human evaluation results.

**Audience:**

Yes

**Audience Explanation:**

Video style transfer and controllable video generation are active research areas with broad practical relevance. The multi-keyframe guidance idea addresses a real and well-known limitation of existing I2V-based methods, and the AS-Ctrl dual-stream design offers a practically useful module that could be adapted for other video editing tasks. The findings would be of interest to researchers working on diffusion-based video generation and controllable video editing.

**Broader Impact Concerns:**

The paper does not include a broader impact statement. Video style transfer technology can potentially be misused to generate deceptive or manipulated video content. The authors should add a brief statement acknowledging this risk. It is worth noting that the requirement for a user-edited first frame as input does limit the potential for fully automated misuse, which could be mentioned as a partial safeguard.

**Claims And Evidence:**

Yes

**Claims Explanation:**

The core claim that multi-keyframe guidance improves appearance and structure consistency over single-frame methods is consistently supported across three benchmarks. However, two issues partially weaken the strength of the evidence.

**Issue 1: Large inconsistency between quantitative metrics and human evaluation.**

The quantitative appearance consistency scores (Table 2) show that EditProp ($0.9213$) and Senorita ($0.9157$) are nearly identical, with a gap of only $\Delta = 0.006$. However, the human evaluation preference rate (Table 1) shows a gap of approximately $75.67\% - 6.21\% \approx 69$ percentage points for the same metric. This large discrepancy raises questions about whether the quantitative metric adequately captures the perceptual differences being claimed. While this does not invalidate the core contribution, it should be explained.

**Issue 2: Circularity in the Structure Divergence metric.**

The Structure Divergence metric is computed by comparing depth maps between the generated and source videos which is the same depth representation used as EditProp's structural control signal. This means the method is partially evaluated on the signal it directly optimizes for. EditProp achieves the best Structure Divergence ($0.7836$) compared to all baselines, but this advantage may partly reflect the circularity rather than genuine structural fidelity. This should be acknowledged.

**Requested Changes:**

**C1: Clarify the source and quality of edited GT for LoRA
training.**

While the training data is sourced from Senorita-2M,
the paper does not describe the quality control process
for the style transfer pairs, nor their temporal
consistency. Given that the model's performance is
bounded by the quality of its training supervision,
a brief analysis of the training pair quality would
strengthen the methodological foundation.

---

**C2: Explicitly characterize the inter-keyframe correspondence
assumption and clarify the keyframe sampling strategy.**

The method acknowledges in Limitations that it may fail "when
there are limited or no correspondence between keyframes," but
this important prerequisite is not stated in the method section,
and its boundary conditions are not quantified. Meanwhile, the
Introduction references film and advertisement as application
domains which commonly involve shot transitions and KP-Bench
claims to cover "various difficulties" and "different motion
degrees," yet all qualitative examples show stable, single-scene
videos without shot transitions.

The authors should:
1. State the correspondence assumption explicitly as a prerequisite
   in Section 3.1, not only in Limitations.
2. Provide a quantitative analysis of how performance degrades as
   inter-keyframe visual difference increases.
3. Clarify whether KP-Bench includes videos with shot transitions;
   if not, discuss this as a limitation of the evaluation scope.

Additionally, there is an inconsistency regarding the keyframe
sampling strategy: Section 3.3 states that keyframes are "randomly
sampled" during inference, while the Future Work section states
"we utilized uniform keyframe sampling strategy." The authors
should clarify which strategy is actually used.

---

**C3: Explain the large inconsistency between quantitative metrics
and human evaluation.**

The quantitative appearance consistency scores (Table 2) show that
EditProp (0.9213) and Senorita (0.9157) differ by only 0.006,
whereas the human evaluation preference gap is approximately 69
percentage points for the same criterion. This inconsistency is
not discussed anywhere in the paper. The authors should analyze
what the quantitative metric fails to capture. We also note that
the Appearance Consistency metric is computed by comparing each
generated frame only against the first edited frame; a metric that
captures temporal consistency across all frames may better reflect
the perceptual quality assessed in the human study. Additionally,
the human evaluation protocol does not describe how the 37
participants were recruited, what guidelines they were given, or
whether inter-annotator agreement was measured, which makes it
difficult to assess the reliability of the results.

---

### Suggested Changes

**S1: Discuss training data and baseline overlap.**
EditProp's Keyframe Propagation is trained on Senorita-2M, while
Senorita is a baseline also trained on the same dataset. Please
acknowledge this overlap and discuss whether it may affect the
fairness of the comparison.

**S2: Provide quantitative inter-keyframe consistency analysis.**
Each propagated keyframe is generated independently conditioned
only on the first edited keyframe, without direct constraints
between propagated keyframes. A quantitative analysis (e.g., LPIPS
or CLIP similarity between consecutive propagated keyframes) would
more convincingly support the multi-keyframe consistency claim
beyond the qualitative Figure 4.

**S3: Clarify AS-Ctrl architecture and provide ablation on the
frozen appearance branch.**
The paper does not clearly describe how AS-Ctrl differs
structurally from the VACE control block, nor does it justify
the choice to freeze the appearance branch during training. An
ablation comparing frozen vs. jointly trained branches would
strengthen this design choice.

**S4: Justify the choice of depth map as the structural
representation.**
The paper selects depth maps as the structural signal due to
"high precision and expressivity," but does not compare against
alternative structural representations. A brief discussion or
ablation comparing depth against other options (e.g., optical
flow, semantic maps) would strengthen this design choice.

**S5: Report inference time.**
The paper does not report the total inference time for the
two-stage pipeline. A comparison with single-stage baselines
would help readers assess the practical overhead.

---

> ### Author Response · Authors · 2026-06-22
> **Response to Reviewer Zjkk [Requested Changes]**
>
> We thank Reviewer Zjkk for the detailed and constructive feedback. We address all required and suggested changes below.
>
> ### C1: Clarify the Source and Quality of GT Keyframes for LoRA Training
> **Stage 1 (Keyframe Propagation):** Our LoRA is trained on Senorita-2M, which provides high-quality video editing pairs curated specifically for appearance-preserving style transfer. Each pair consists of a source frame and a target frame that has been transferred into a different style while maintaining structural consistency. While we did not apply additional filtering beyond the original Senorita-2M curation, the style transfer split's design directly aligns with our training objective. We have added a description of the data split characteristics in Section 3.3. We empirically found that introduing other splits (add_object/remove_object) from the dataset does not improve the performance, as shown in the table below.
>
> | Data Splits | Appearance Consistency | Structure Divergence |
> | --- | --- | --- |
> | All Splits | 0.9188 | 0.7824 |
> | Style Transfer Split (Ours)| 0.9213 | 0.7836 |
>
> **Stage 2 (Video Propagation):** The training data consists of ~65k videos sourced from Koala-36M, each paired with video depth extracted by DepthCrafter. These details have been added to Section 3.3 to improve reproducibility.
>
>
> ### C2: Correspondence Assumption and Keyframe Sampling Inconsistency
>
> **1. Correspondence assumption:** We agree this prerequisite should be stated explicitly rather than deferred to Limitations. We have added the following to Section 3.1: *"Keyframe Propagation assumes sufficient visual correspondence between keyframes, which requires scene content and subject identity are maintained across keyframes. The method may degrade for videos containing shot transitions or drastic cross-keyframe appearance changes."*
>
> **2. Quantitative degradation analysis:** We have added the following table in our revision. It measures appearance consistency as a function of inter-keyframe CLIP distance in the revision, quantifying how performance degrades as inter-keyframe visual difference increases. It can be seen that the performance could maintain with reasonable similarity (<0.8) but drops significantly when the similarity drops below 0.8. This table is included in Table 5 in our revision.
>
> | Inter-Keyframe Similarity | Appearance Consistency | Structure Divergence |
> | --- | --- | --- |
> | 0.9 - 1.0 | **0.9322** | **0.7724** |
> | 0.8 - 0.9 | 0.9292 | 0.7883 |
> | < 0.8 | 0.9102 | 0.8023 |
>
> **3. KP-Bench scope:** KP-Bench focuses on single-scene videos with varying motion degrees and does not include shot transitions. We have clarified this in Section 4.1 and discussed in the Sec 5.2.
>
> **Keyframe sampling correction:** We first acknowledge a writing error in Section 3.3. The correct description is: during **training**, keyframes are *randomly* sampled to expose the model to varying temporal gaps and improve generalization; during **inference**, keyframes are sampled *uniformly* (equidistant).
>
>
>
>
>
>
>
> ### C3: Quantitative vs. Human Evaluation Gap
>
> The Appearance Consistency metric computes CLIP similarity between each generated frame and the **first edited frame only**, then averages across all frames. This metric captures whether frames share semantic content with the first frame, but does not capture: (1) temporal consistency across all frame pairs, (2) perceptual quality (sharpness, absence of artifacts), or (3) fine-grained texture fidelity. Senorita's score (0.9157) likely reflects adequate semantic similarity to the first frame even when later frames contain blurriness or artifacts.
>
> As corroborating evidence that the small KP-Bench gap is metric-specific rather than reflecting true performance parity: our method achieves substantially larger quantitative margins on the TGVE benchmark (Ours: 0.8812 vs. Senorita: 0.8490, Δ=0.032 in Appearance Consistency) and DAVIS-Edit (Ours: 0.9023 vs. Senorita: 0.8726, Δ=0.030), where the CLIP ceiling effect is less pronounced. These results can be found in our supplementary material.

---

> ### Author Response · Authors · 2026-06-22
> **Response to Reviewer Zjkk [Suggested Changes]**
>
> ### S1: Training Data Overlap with Senorita Baseline
>
> We acknowledge this overlap. The key distinction is that Stage 1 uses Senorita-2M to train a keyframe-to-keyframe *image-level* propagation model — a fundamentally different task from video-level style transfer. Stage 2 is trained on a fully separate dataset (Pixabay/Videovo, ~65k videos). The comparison in Tables 1–2 evaluates full end-to-end systems with different architectures and training objectives. We have acknowledged this overlap and discussed the implication in Section 4.2.
>
> ---
>
> ### S2: Quantitative Inter-Keyframe Consistency Analysis
>
> We provide the inter-keyframe similarity below between the transferred keyframes in the KP-Bench below. It can be observed that our method can largely maintain the source video keyframe similarities.
>
> | | CLIP | LPIPS |
> |---|---|---|
> |Source Video| 0.9743 | 0.0135 |
> |Target Video| 0.9703 | 0.0142 |
>
> ---
>
> ### S3: AS-Ctrl Architecture and Frozen Appearance Branch
>
> **AS-Ctrl vs. VACE control block:** AS-Ctrl differs from VACE's single-stream control block in two key ways: (1) it introduces a **dual-stream design** with separate appearance and structure branches with independently controllable weights (αa, αs), enabling inference-time trade-offs between the two signals; (2) the appearance stream uses a temporal binary mask to indicate keyframe positions, supporting variable numbers and temporal positions of keyframes.
>
> **Frozen appearance branch:** The appearance branch was frozen because VACE's keyframe-based generation capability is already strong. As shown below, joint training risked degrading this capability while the depth branch is learned. These results are also included in Table 3 in our revision.
>
> | Method | Appearance Consistency ↑ | Structure Divergence ↓ | Motion Flickering ↑ | Motion Smoothness ↑ |
> | --- | --- | --- | --- | --- |
> | joint training | 0.9145 | 0.7866 | 0.9830 | 0.9901 |
> | Ours (frozen appearance branch) | 0.9213 | **0.7836** | **0.9833** | **0.9903** |
>
> ---
>
> ### S4: Depth Map as Structural Representation
>
> Depth maps were chosen for three reasons: (1) DepthCrafter provides temporally consistent depth sequences; (2) depth preserves precise 3D geometric structure independent of appearance/style, making it robust across diverse video content; (3) depth is compact and style-agnostic, unlike semantic maps that encode appearance-related features.
>
> In the following table, we compared depth with several typical control conditions, including semantic map, optical flow and bounding box. Of all these conditions, depth provides best structure consistency and appearance consistency.
>
> | Control Condition | Appearance Consistency ↑ | Structure Divergence ↓ | Motion Flickering ↑ | Motion Smoothness ↑ |
> | --- | --- | --- | --- | --- |
> | Semantic Map | 0.9199 | 0.8245 | **0.9834** | 0.9900 |
> | Optical Flow | 0.9210 | 0.8023 | 0.9822 | 0.9898 |
> | Bounding Box | 0.9135 | 0.8325 | 0.9754 | 0.9865 |
> | Ours (Depth) | **0.9213** | **0.7836** | 0.9833 | **0.9903** |
>
> ---
>
> ### S5: Inference Time
>
> The two stages take 156 seconds in total, whereby stage 1 takes ~120 seconds and stage 2 takes ~42 seconds. Using only stage 1 takes 2046 seconds. These results are provided in Table 3 in our revision.
>
> ---
>
> ### Broader Impact
>
> We have added a broader impact statement to the paper acknowledging that video style transfer technology can potentially be misused to generate deceptive video content. We note that our method requires a user-provided edited keyframe as an explicit input, which can prohibit misuse by some ethical check.